# Immunosuppression with Calcineurin Inhibitor after Renal Transplant Failure Inhibits Allosensitization

**DOI:** 10.3390/biomedicines8040072

**Published:** 2020-03-28

**Authors:** Covadonga López del Moral Cuesta, Sandra Guiral Foz, David Gómez Pereda, José Luis Pérez Canga, Marina de Cos Gómez, Jaime Mazón Ruiz, Ana García Santiago, José Iñigo Romón Alonso, Rosalía Valero San Cecilio, Emilio Rodrigo Calabia, David San Segundo Arribas, Marcos López Hoyos, Juan Carlos Ruiz San Millán

**Affiliations:** 1Nephrology Department, University Hospital Marqués de Valdecilla-IDIVAL, 39008 Santander, Spain; davidgomezpereda@hotmail.es (D.G.P.); joseluis.perezc@scsalud.es (J.L.P.C.); marina.decos@scsalud.es (M.d.C.G.); jaime.mazon@scsalud.es (J.M.R.); ana.garcias@scsalud.es (A.G.S.); rosalia.valero@scsalud.es (R.V.S.C.); emilio.rodrigo@scsalud.es (E.R.C.); juancarlos.ruiz@scsalud.es (J.C.R.S.M.); 2Immunology Department, University Hospital Marqués de Valdecilla-IDIVAL, 39008 Santander, Spain; sandraalmudena.guiral@scsalud.es (S.G.F.); david.sansegundo@scsalud.es (D.S.S.A.); marcos.lopez@scsalud.es (M.L.H.); 3Hematology Department, University Hospital Marqués de Valdecilla-IDIVAL, 39008 Santander, Spain; joseinigo.romon@scsalud.es

**Keywords:** allosensitization, donor-specific antibody, calcineurin inhibitor, graft nephrectomy

## Abstract

Immunosuppression withdrawal after graft failure seems to favor sensitization. A high percentage of calculated panel-reactive antibody (cPRA) and the development of de novo donor specific antibodies (dnDSA) indicate human leukocyte antigen (HLA) sensitization and may hinder the option of retransplantation. There are no established protocols on the immunosuppressive treatment that should be maintained after transplant failure. A retrospective analysis including 77 patients who lost their first renal graft between 1 January 2006–31 December 2015 was performed. Two sera were selected per patient, one immediately prior to graft loss and another one after graft failure. cPRA was calculated by Single Antigen in all patients. It was possible to analyze the development of dnDSA in 73 patients. By multivariate logistic regression analysis, the absence of calcineurin inhibitor (CNI) at 6 months after graft failure was related to cPRA > 75% (OR 4.8, CI 95% 1.5–15.0, *p* = 0.006). The absence of calcineurin inhibitor (CNI) at 6 months after graft loss was significantly associated with dnDSA development (OR 23.2, CI 95% 5.3–100.6, *p* < 0.001). Our results suggest that the absence of CNI at the sixth month after graft loss is a risk factor for sensitization. Therefore, maintenance of an immunosuppressive regimen based on CNI after transplant failure should be considered when a new transplant is planned, since it seems to prevent HLA allosensitization.

## 1. Introduction

The overall survival of renal allografts has improved over the past decade. Despite obtaining better results in the short term, the rate of long-term renal graft loss remains stable [1,2]. It is estimated that approximately 15% of kidney transplant recipients lose their graft in the first 5 years after transplantation [3]. Patients with graft failure represents 4% of the population on hemodialysis, and approximately 15–20% of patients on the waiting list for retransplantation [4].

Patients with graft failure constitute a group with increased morbidity and mortality [5]. The first year in dialysis increases mortality, and the mortality rate is higher in patients who return to dialysis after transplant failure than in those patients who have not received any kidney transplant. After the first year, mortality drops, but remains higher than in transplant-naive patients [6,7,8]. Due to this, the possibility of performing a retransplant is crucial in these patients, and this option significantly reduces mortality in approximately 45% [9,10]. Graft survival after retransplantation is lower than after the first graft, but remains sufficiently high [11,12].

The possibility of receiving a new transplant is influenced by human leukocyte antigen (HLA) sensitization, and graft survival may be reduced if there are preexisting donor specific antibodies (DSA) [13]. Previous transplants, pregnancies and blood transfusions are the major causes of sensitization. HLA sensitization risk factors after transplant failure are, among others, immunosuppression (IS) weaning, blood transfusions [14,15,16] and the performance of graft nephrectomy.

Graft nephrectomy is a known risk factor for HLA allosensitization, although its underlying mechanisms are not well established [17,18]. Graft nephrectomy is associated with increased morbidity and mortality and loss of residual renal function [19]. The graft that remains in situ can produce chronic inflammation, resulting in elevated C-reactive protein, hypoalbuminemia and malnutrition [20]. It is recognized that de novo DSA (dnDSA) can be developed after nephrectomy, and it seems that the inflammation state from endothelial damage at the time of nephrectomy can induce the formation of DSA. Although the mechanisms by which this HLA sensitization occurs are not totally clear, nephrectomy can make the chance of receiving a further transplant less likely [21,22,23,24].

On the other hand, immunosuppression weaning seems a predictor of sensitization. After transplant failure, it is not well established which IS strategy should be maintained [25]. It seems that the maintenance of IS could prevent the development of anti-HLA antibodies. The risk of antibody formation and sensitization must be balanced with the long-term risks of infectious processes, malignant neoplasms and metabolic alterations. Therefore, HLA sensitization in the context of IS withdrawal may hinder retransplantation.

Few studies have analyzed the relationship between immunosuppression withdrawal and the development of anti-HLA antibodies, and there is no clear recommendation on how to manage immunosuppressive treatment after graft failure [26,27,28,29,30,31]. Nimmo et al. performed a single-center retrospective study with 41 patients with transplant failure, measuring DSA before IS wean, after IS wean and after IS withdrawal. DSAs were also measured before and after graft nephrectomy in those in which it was performed. An increase in antibody titer was observed in those patients with IS withdrawal, and this titer was higher in patients with graft nephrectomy [32].

Recently, Lucisano et al. have conducted a single-center study with 109 patients (61 patients with graft nephrectomy; 48 patients without graft nephrectomy). Antibody development was evaluated up to 24 months after graft failure. The group of patients with graft nephrectomy had a higher antibody titer, being considered an independent risk factor for the development of DSA. In patients without graft nephrectomy, low levels of tacrolimus were related to the formation of DSA, and it was observed that tacrolimus levels ≥ 3 ng / ml were protective against allosensitization [33].

Despite this evidence, there is still no clear recommendation on the optimal management of IS in patients returning to dialysis after transplant failure. This single-center retrospective study aims to demonstrate whether a specific maintenance immunosuppression regimen after graft failure prevents HLA allosensitization.

## 2. Materials and Methods

A retrospective analysis including those patients who lost their first renal graft between 1 January 2006–31 December 2015 at University Hospital Marqués de Valdecilla was performed. In total, 106 patients with death with functioning graft were excluded and 70 patients in whom serum was not available before or/and after graft failure were also excluded. Thus, 22 patients with preemptive second kidney transplant were included (Figure 1). We have got the written informed consent from the patients, and the study was conducted according to the guidelines of the Declaration of Helsinki and was approved by the Hospital’s Ethics Committee (31 Jan 2020, Project identification code 2019.325).

Patients were divided into two groups according to maintenance immunosuppression. Group 1 (“No-CNI 3mo”) was made up of patients with no immunosuppressive treatment, or only with maintenance of corticosteroids or mycophenolate mofetil (MMF) / mycophenolic acid (MPA) until at least the third month after transplant failure. Group 2 patients were those with maintenance of calcineurin inhibitor (CNI) until at least the third month after graft failure, with or without MMF/MPA/corticosteroids. In this group (“CNI 3mo”) some patients (*n* = 22) who received a second preemptive kidney transplant were included. Similarly, patients were classified according to the presence or not of CNI until at least the sixth month after transplant failure. Group A (“No-CNI 6mo”) was made up of patients with no sixth month CNI maintenance and conversely, group B (“CNI 6mo”) was made up of patients with CNI maintenance at the sixth month after graft loss. In group B, patients who received a second preemptive renal transplant were included.

As routine clinical practice in our hospital, serum samples were collected and stored every 3 months in patients on the waiting list for retransplantation. In kidney transplant recipients, serum samples were collected every 3–4 months. In our study, two sera were selected per patient, one immediately prior to graft loss and another one after graft failure. In patients with preemptive second transplant, the serum selected after first graft failure was 6–12 months after the second renal transplant. In those patients with nonpreemptive second transplant, the serum selected after first graft failure was prior to the performance of second kidney transplant. In patients in whom transplantectomy was performed, the selected serum after transplant failure was already after graft nephrectomy. Donor (donor type, age), transplant (cold ischemia time, graft nephrectomy) and recipient data (age, comorbidities, cause of chronic kidney disease) were collected from the prospectively maintained database of renal transplant patients at our center. In addition, the number of blood transfusions per patient was recorded from transplant failure to the date of serum after graft loss.

Immunosuppressive treatment data were collected at the time of serum before transplant failure. Maintenance immunosuppression was also collected in months 1, 3 and 6 after graft failure. 

Anti-HLA antibodies were analyzed in selected sera by Single Antigen (LABSCreen, Single Antigen class-I and class-II, One Lambda, Canoga Park, CA). Calculated panel-reactive antibody (cPRA) provides an estimate of the percentage of deceased organ donors that will be crossmatch incompatible for a candidate, and it was obtained through Virtual PRA Calculator of the Eurotransplant Reference Laboratory (https://www.etrl.org/Virtual%20PRA/). The difference between cPRA in sera selected per patient before and after transplant failure (delta cPRA), and the formation of de novo donor-specific anti-HLA - A, B, C, DR and DQ - antibodies (dnDSA) after graft loss were also calculated. 

The relationship of different variables with the development of cPRA > 75% after graft failure and delta cPRA > 0% was analyzed. These values were taken since they were close to the median value and, therefore, divided the patients into two similar groups. Considering that some patients were already sensitized prior to graft failure, previously non-sensitized patients were considered to analyze cPRA > 75% after graft loss. The development of de novo DSA was analyzed in those patients in which donor typing was available.

### Statistical Analysis

Continuous variables were expressed as mean ± standard deviation (SD) or median and interquartile range according to their distribution. Categorical variables were described as relative frequencies. Averages were compared using Student’s t-test for continuous variables with normal distribution, and non-parametric tests (Mann-Whitney U test) for those with non-normal distribution. Chi-square test was used to compare averages of categorical variables. The relationship between cPRA > 75% after graft loss and different variables such as graft nephrectomy and CNI withdrawal was analyzed by logistic regression. Similarly, the relationship between delta cPRA > 0% and de novo DSA with different variables was analyzed by logistic regression. *p* values less than 5% defined statistical significance. Statistical analysis was carried out using the SPSS statistical software package, version 22.0 (SPSS Inc, Chicago, IL, USA).

## 3. Results

Seventy-seven patients who lost their first renal graft were included. The mean age of the patients at the time of graft failure was 60.4 ± 12.8 years, and 67.5% were male. In total, 94.8% of the patients had hypertension, and 32.5% had diabetes mellitus. The median duration of the first renal graft was 10.0 years (interquartile range 6.0–14.0). Graft nephrectomy was performed in 23.0% of the patients. 65 of the patients (84.4%) received a second kidney transplant, and in 22 of them a preemptive second transplant was performed.

All patients with CNI had tacrolimus as maintenance immunosuppressive treatment, except 2 patients who had cyclosporine. Antimetabolite treatment was based on MMF or MPA. CNI and/or corticosteroids was the immunosuppression of choice in most patients who had treatment after transplant failure. Corticosteroid maintenance, in monotherapy or in combination with other immunosuppressive drugs, at the third (*p* = 0.247) and sixth month (*p* = 0.322) after graft loss was not significantly associated with cPRA ≤ 75% in all patients, and in previously non-sensitized patients (*p* = 0.429 and 0.514, respectively). Similarly, the maintenance of corticosteroids at the third and sixth month was not associated with delta cPRA ≤ 0% (*p* = 0.197 and 0.129) or no de novo DSA formation (*p* = 0.108 and 0.233). For all the above, patients were classified according to the maintenance or not of immunosuppressive treatment with CNI after graft failure.

### 3.1. Characteristics of Patient Groups

In group 1 (34 patients, 44.2%), in which patients without CNI were included, 17 patients (50.0%) had no immunosuppressive treatment after transplant failure, 16 patients (47.0%) maintained only corticosteroids and 1 patient (2.9%) maintained MMF/MPA alone or in combination with corticosteroids. In group 2 (43 patients, 55.8%), which was made up of patients with CNI at least until the third month after transplant failure, 10 patients (23.2%) had immunosuppressive treatment with CNI in monotherapy until at least the third month after graft loss and 33 patients (76.7%) with CNI and MPA/MMF and/or corticosteroids. In relation to maintenance immunosuppressive treatment at 6 months after graft failure, 53.4% of patients did not have CNI at the sixth month (group A). In this group, 23 patients (56.0%) had no immunosuppressive treatment after transplant failure, 17 patients (41.4%) maintained only corticosteroids and 1 patient (2.4%) maintained MMF/MPA alone or in combination with corticosteroids. In group B (36 patients, 46.6%), which was made up of patients with CNI at least until the sixth month after transplant failure, 5 patients (13.8%) had immunosuppressive treatment with CNI in monotherapy, and 31 patients (86.1%) had CNI with MPA/MMF and/or corticosteroids.

### 3.2. Selected Sera and Sensitized Patients before Transplant Failure

Selected sera were extracted 1.0 month before graft failure (interquartile range 0.3–2.1) and 7.6 months after transplant failure date (interquartile range 5.9–12.5). Median cPRA before graft loss was 0.0% (interquartile range 0.0–0.0). Thus, 79.2% of the patients were not sensitized before transplant failure (cPRA 0.0%) and 88.3% of the patients had cPRA ≤ 75% before graft failure.

### 3.3. Development of cPRA > 75% after Transplant Failure

It was observed that 46.7% of the patients showed cPRA > 75% after transplant failure (Table 1). Both graft nephrectomy and group 1 (“No-CNI 3mo”) were associated with cPRA > 75% significantly. Similarly, group A (“No-CNI 6mo”) was associated with cPRA > 75% after transplant failure, as shown in Figure 2. Conversely, group 2 (“CNI 3mo”) and group B (“CNI 6mo”) patients were significantly associated with cPRA ≤ 75%. By multivariate logistic regression analysis, graft nephrectomy was an independent risk factor for cPRA > 75% after transplant failure. Patients without CNI maintenance at the third and sixth month after graft loss (group 1 and group A, respectively) were related to cPRA > 75% in the multivariate analysis. The results of the multivariate analysis for cPRA > 75% are shown in Table 2. Considering only previously non-sensitized patients (79.2%), 45.9% showed cPRA > 75% after graft failure. Similarly, group 1 (“No-CNI 3mo”) and group A (“No-CNI 6mo”) were significantly associated with cPRA > 75% in the multivariate analysis. In this group of patients, graft nephrectomy was also associated with cPRA > 75%, but this result was not statistically significant by multivariate logistic regression analysis. These results are shown in Table 3 and Table 4.

### 3.4. Development of Delta cPRA > 0% after Transplant Failure

In total, 49.3% of the patients presented delta cPRA > 0% (Table 5). A shorter duration of the first renal graft was associated with delta cPRA > 0% significantly. Graft nephrectomy was significantly associated with delta cPRA > 0%. Similarly, patients without CNI maintenance at the third and sixth month after graft loss (group 1 and group A, respectively) were related to delta cPRA > 0%. Conversely, group 2 (“CNI 3mo”) and group B (“CNI 6mo”) were significantly associated with delta cPRA ≤ 0%. Table 6 shows the results of the multivariate logistic regression analysis for delta cPRA > 0%. Transplantectomy was an independent risk factor for delta cPRA > 0%. As logistic regression analysis with cPRA > 75%, group 1 and group A patients were associated with delta cPRA > 0% in the multivariate analysis.

### 3.5. De Novo DSA Formation in Patients with Available Donor Typing

It was possible to analyze the development of de novo DSA (anti-HLA A, B, C, DR and DQ) after transplant failure in 73 patients, due to the lack of donor typing data in the rest of the patients included in our study. In actuality, 45.2% of patients developed dnDSA. Graft nephrectomy and patients without CNI maintenance at the third month after graft loss (group 1) were significantly associated with dnDSA formation. Group A patients were also associated with the development of dnDSA significantly (Figure 3). Conversely, group 2 and group B patients were significantly associated with no dnDSA formation. The duration of the first renal graft was shorter in those patients with development of dnDSA. These results are shown in Table 7. By multivariate logistic regression analysis (Table 8), patients without CNI maintenance at the third and sixth month after graft loss (group 1 and group A, respectively) were associated with dnDSA formation.

## 4. Discussion

Despite the improvement in renal transplant results, the number of patients with long-term graft loss remains significant [1,2,3]. Patients who return to dialysis after graft failure have a high mortality [4,5]. Retransplantation is an important option to consider, since there is a mortality reduction around 45% in these patients [9,10,11,12]. HLA allosensitization is a determining factor that prolongs the waiting time for transplantation. The most important causes of HLA sensitization are previous transplants, pregnancies and blood transfusions. After graft failure, transplantectomy and IS withdrawal have been related to HLA sensitization.

Graft nephrectomy seems to favor the development of anti-HLA antibodies. The exact mechanisms are not clear, but it is known that graft nephrectomy induces dnDSA formation due to endothelial damage [23,24]. Recently, Lucisano et al. [33] showed that graft nephrectomy is followed by the long-term production of DSA and non-DSA HLA antibodies. Moreover, this study showed that transplantectomy is an independent risk factor for developing DSA at 12 and 24 months after graft failure (*p* = 0.005 and 0.008). Similarly, our study shows that graft nephrectomy favors the development of anti-HLA antibodies after transplant failure. By multivariate logistic regression analysis, graft nephrectomy is an independent factor that increases the risk of developing cPRA > 75% after transplant failure and delta cPRA > 0%. Transplantectomy is also associated with dnDSA formation, but this result is not statistically significant in the multivariate analysis.

Blood transfusions are sensitizing events that favor the development of anti-HLA antibodies. This HLA allosensitization after transfusions could also hinder the retransplant option [14,15,16]. In our patients, 49.4% have at least one blood transfusion after graft failure until serum extraction, with a median transfusion of 1.0 red blood cells concentrates (interquartile range 0.0–4.0). However, by dividing patients into groups according to the percentage of cPRA after transplant failure or delta cPRA, there are no statistically significant differences in relation to having received at least one blood transfusion after transplant failure.

The main finding of our study is that CNI maintenance after graft loss reduces the development of anti-HLA antibodies, avoiding new sensitization. The management of IS after transplant failure is a controversial topic and it is questioned in different studies. It seems that the complete IS withdrawal after transplant failure favors the development of anti-HLA antibodies, but there is no clear evidence since there are few studies that have analyzed it. Currently, there are no established protocols and the option of maintaining or not immunosuppressive treatment depends on the physician’s choice [25,26,27,28,29,30,31]. In addition, there is an increased risk of infectious complications, malignancies or metabolic complications in patients in whom immunosuppressive therapy is maintained. Nimmo et al. [32] observed an increase in antibody titer in those patients with IS withdrawal after transplant failure. Lucisano et al. [33] described that in patients with a graft *in situ*, the maintenance of tacrolimus with levels ≥ 3 ng/mL prevents the development of anti-HLA antibodies. In our study, the absence of CNI at 6 months after transplant failure is an independent factor that increases the risk of cPRA > 75% more than four times in previously non-sensitized patients. Similarly, no CNI maintenance at 6 months after graft failure increases the risk of delta cPRA > 0% more than five times in all our patients. Moreover, the absence of immunosuppressive maintenance treatment with CNI at 6 months after graft loss is a strong, independent risk factor for the development of dnDSA.

As the immunosuppressive treatment after graft loss is not established, our patients have different immunosuppression patterns. In our study, patients are divided into groups according to maintenance immunosuppression until the third or sixth month after transplant failure. Group 1 (“No-CNI 3mo”) and group A (“No-CNI 6mo”) are associated with a higher risk of cPRA > 75% after graft failure and delta cPRA > 0% in the multivariate analysis. Similarly, in those patients in whom it was possible to analyze whether they developed dnDSA, it is observed that group 1 and group A patients are significantly associated with dnDSA formation. In group 2 (“CNI 3mo”) and group B (“CNI 6mo”) are included those patients with preemptive second transplantation. Therefore, these patients, despite having a sensitizing event such as receiving a second transplant, have protection against HLA sensitization. This result could probably be explained by the absence of IS withdrawal in these patients.

The strength of our study is essentially to have two sera available per patient, one before first graft loss and the other one after transplant failure. This makes possible to determine not only cPRA after graft failure but also delta cPRA between the two sera. In addition, this allows to analyze dnDSA development from graft failure, since patients first serum is immediately prior to transplant failure.

Despite these results, our study presents several limitations. This is a retrospective study, and there are no sera available in all patients after transplant failure. Patients without collected sera after graft loss are probably those in which retransplantation is unlikely. Moreover, the patients in our study have different immunosuppression patterns. The groups without CNI (groups 1 and A) include some patients with other immunosuppressive drugs, but this probably enhances the result in favor of CNI in the prevention of sensitization after graft failure. Otherwise, no levels of CNI are performed in our patients after graft failure, which does not allow to establish protection levels of CNI to prevent allosensitization. Finally, it is not possible to obtain dnDSA in all our patients since donor typing is not completely available.

With our results we can conclude that the absence of maintenance treatment with CNI at the sixth month after graft loss is a risk factor for sensitization. Therefore, immunosuppressive maintenance treatment based on CNI after transplant failure should be considered when a new transplant is planned in the short-medium term and in the absence of a high antibody titer, since it seems to prevent HLA allosensitization. Multicenter prospective studies are required.

## Figures and Tables

**Figure 1 biomedicines-08-00072-f001:**
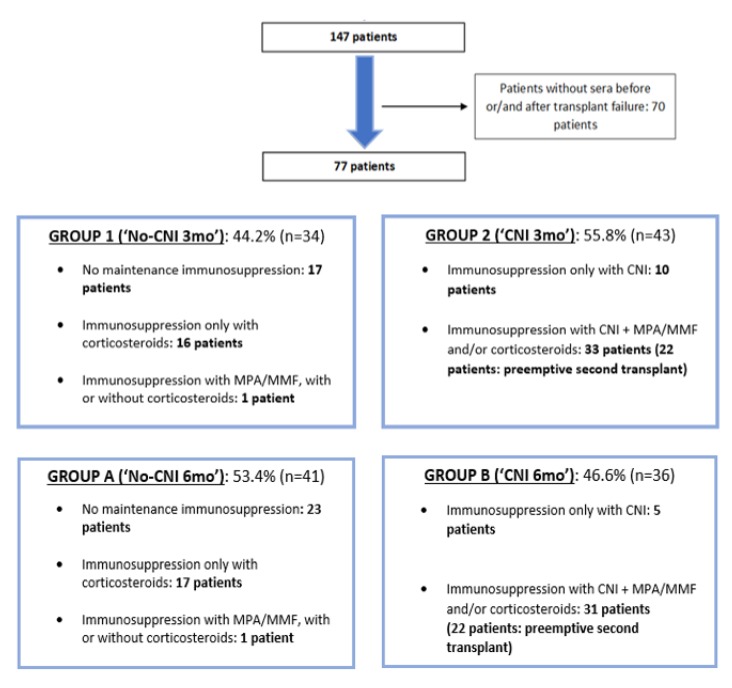
Subject selection and patient groups according to maintenance immunosuppression until the third (group 1 and 2) and sixth month (group A and B) after transplant failure.

**Figure 2 biomedicines-08-00072-f002:**
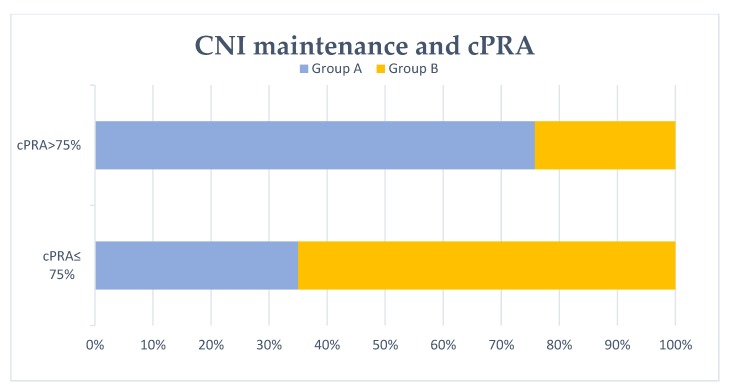
Patients with/without calcineurin inhibitor (CNI) maintenance at 6 months and cPRA after transplant failure. No CNI maintenance at 6 months after graft failure (group A) was associated with cPRA >75% significantly (*p* = 0.001).

**Figure 3 biomedicines-08-00072-f003:**
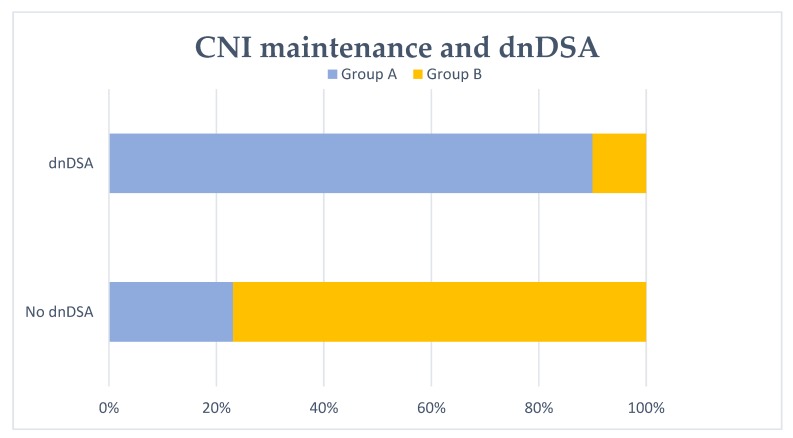
De novo DSA (dnDSA) development and no CNI maintenance at 6 months. No CNI maintenance at 6 months after transplant failure (group A) was significantly associated with dnDSA formation (*p* < 0.001).

**Table 1 biomedicines-08-00072-t001:** Patient characteristics in relation to the development of calculated panel-reactive antibody (cPRA) ≤ or >75% after transplant failure. Continuous variables were expressed as mean ± SD (*) or median and interquartile range (^) according to their distribution. The bold: *p* values less than 0.05 defined statistical significance.

cPRA and Different Variables	*n* = 77	cPRA ≤ 75% *n* = 41	cPRA > 75% *n* = 36	*p*
Recipient age (years) *	60.4 ± 12.8	60.6 ± 12.8	59.6 ± 13.1	0.740
Recipient sex (male)	67.5%	68.3%	66.7%	0.879
Cause of chronic kidney disease (CKD)	-	-	-	0.163
Vascular	7.8%	7.3%	8.3%	
Diabetes	14.3%	7.3%	22.2%	
Others	77.9%	85.4%	69.4%	
HTN	94.8%	95.1%	94.4%	0.894
Diabetes	32.5%	29.7%	43.8%	0.227
RRT before first transplant:	-	-	-	-
HD	62.3%	70.6%	68.6%	0.856
PD	27.3%	26.5%	34.3%	0.481
Donor age (years) *	48.3 ± 18.3	49.0 ± 17.8	47.5 ± 19.0	0.723
Type of donor	-	-	-	0.098
Deceased donor	96.1%	92.7%	100.0%	
Living donor	3.9%	7.3%	0.0%	
Cause of death (deceased donor)	-	-	-	0.141
DBD	97.3%	100.0%	94.4%	
DCD	2.7%	0.0%	5.6%	
Cold ischemia time (hours) ^	20.0 (18.0–24.0)	20.0 (18.0–24.0)	20.0 (18.0–24.0)	0.639
Duration of the first graft (years) ^	10.0 (6.0–14.0)	11.0 (6.5–14.0)	9.5 (5.7–15.2)	0.051
Cause of graft loss	-	-	-	0.245
Chronic allograft nephropathy	75.3%	82.9%	66.7%	
Antibody-mediated rejection	1.3%	2.4%	0.0%	
Recurrence	3.9%	2.4%	5.6%	
Arterial/venous thrombosis	14.3%	9.8%	19.5%	
Others	5.2%	2.5%	8.2%	
Time from graft failure to subsequent serum (months) ^	7.6 (5.9–12.5)	6.9 (5.8–10.3)	8.5 (6.3–16.4)	0.143
Graft nephrectomy	23.0%	7.7%	40.0%	0.001
Blood transfusions from transplant failure to subsequent serum (≥1 RBCC)	49.4%	56.1%	41.7%	0.206
cPRA before transplant failure ^	0.0 (0.0–0.0)	0.0 (0.0–0.0)	0.0 (0.0–0.0)	0.991
Group 1 (“No-CNI 3mo”)	44.2%	24.4%	66.7%	<0.001
Group A (“No-CNI 6mo”)	53.4%	35.0%	75.8%	0.001

**Table 2 biomedicines-08-00072-t002:** Results of the multivariate logistic regression analysis for cPRA > 75%. The bold: *p* values less than 0.05 defined statistical significance.

Multivariate logistic regression analysis for cPRA > 75%	**OR**	**95% CI**		***p***
		**INF**	**SUP**	
Graft nephrectomy	5.1	1.1	23.6	**0.034**
Group 1 (“No-CNI 3mo”)	4.3	1.5	12.8	**0.007**
Recipient age	1.0	0.9	1.0	0.597
	**OR**	**95% CI**		***p***
		**INF**	**SUP**	
Graft nephrectomy	4.9	1.0	22.2	**0.038**
Group A (“No-CNI 6mo”)	4.8	1.5	15.0	**0.006**
Recipient age	1.0	0.9	1.0	0.867

**Table 3 biomedicines-08-00072-t003:** Patient characteristics in relation to the development of cPRA ≤ or >75% after transplant failure in previously non-sensitized patients. Continuous variables were expressed as mean ± SD (*) or median and interquartile range (^) according to their distribution. The bold: *p* values less than 0.05 defined statistical significance.

cPRA and Different Variables in Non-Sensitized Patients	*n* = 61	cPRA ≤75%*n* = 33	cPRA >75%*n* = 28	*p*
Recipient age (years) *	59.9 ± 13.3	61.7 ± 13.0	57.8 ± 13.7	0.252
Recipient sex (male)	70.5%	69.7%	71.4%	0.883
Cause of CKD:	-	-	-	0.112
Vascular	4.9%	6.1%	3.6%	
Diabetes	14.8%	6.1%	25.0%	
Others	80.3%	87.9%	71.4%	
HTN	95.1%	97.0%	92.9%	0.459
Diabetes	33.3%	30.0%	37.5%	0.561
RRT before first transplant:	-	-	-	-
HD	69.1%	70.4%	67.9%	0.840
PD	25.5%	18.5%	32.1%	0.246
Donor age (years) *	49.8 ± 17.2	49.9 ± 17.4	49.6 ± 17.3	0.946
Type of donor	-	-	-	0.185
Deceased donor	96.7%	93.9%	100.0%	
Living donor	3.3%	6.1%	0.0%	
Cause of death (deceased donor)	-	-	-	0.130
DBD	96.6%	100.0%	92.9%	
DCD	3.4%	0.0%	7.1%	
Cold ischemia time (hours) ^	20.0 (18.0–24.0)	20.0 (18.0–23.5)	20.5 (20.0–25.0)	0.290
Duration of the first graft (years) ^	6.5 (1.3–11.7)	9.9 (3.4–13.1)	4.6 (0.0–10.8)	0.093
Cause of graft loss	-	-	-	0.343
Chronic allograft nephropathy	73.8%	81.8%	64.3%	
Antibody-mediated rejection	1.6%	3.0%	0.0%	
Recurrence	3.3%	3.0%	3.6%	
Arterial/venous thrombosis	18.0%	12.2%	25.0%	
Others	3.3%	0.0%	7.1%	
Time from graft failure to subsequent serum (months) ^	7.6 (6.0–12.1)	6.9 (5.9–10.5)	8.4 (6.6–14.6)	0.374
Graft nephrectomy	26.7%	9.4%	46.4%	0.001
Blood transfusions from transplant failure to subsequent serum (≥1 RBCC)	45.9%	51.5%	39.3%	0.340
Group 1 (“No-CNI 3mo”)	49.2%	27.3%	75.0%	<0.001
Group A (“No-CNI 6mo”)	56.1%	37.5%	80.0%	0.001

**Table 4 biomedicines-08-00072-t004:** Results of the multivariate logistic regression analysis for cPRA > 75% in previously non-sensitized patients. The bold: *p* values less than 0.05 defined statistical significance.

Multivariate Logistic Regression Analysis for cPRA > 75% in Non-Sensitized Patients	OR	95% CI		*p*
		INF	SUP	
Graft nephrectomy	4.5	0.9	20.8	0.053
Group 1 (“No-CNI 3mo”)	5.6	1.6	19.3	0.006
Recipient age	0.9	0.9	1.0	0.534
	OR	95% CI		*p*
		INF	SUP	
Graft nephrectomy	4.6	1.0	21.2	0.049
Group A (“No-CNI 6mo”)	4.9	1.3	18.4	0.018
Recipient age	0.9	0.9	1.0	0.389

**Table 5 biomedicines-08-00072-t005:** Patient characteristics in relation to the development of delta cPRA ≤ or >0% after transplant failure. Continuous variables were expressed as mean ± SD (*) or median and interquartile range (^) according to their distribution. The bold: *p* values less than 0.05 defined statistical significance.

Delta cPRA and Different Variables	*n* = 77	Delta cPRA ≤ 0% *n* = 39	Delta cPRA > 0% *n* = 38	*p*
Recipient age (years) *	60.4 ± 12.8	60.3 ± 12.9	60.0 ± 13.0	0.911
Recipient sex (male)	67.5%	66.7%	68.4%	0.869
Cause of CKD:	-	-	-	0.585
Vascular	7.8%	7.7%	7.9%	
Diabetes	14.3%	10.3%	18.4%	
Others	77.9%	82.1%	73.7%	
HTN	94.8%	97.4%	92.1%	0.292
Diabetes	32.5%	37.1%	35.3%	0.873
RRT before first transplant:	-	-	-	-
HD	62.3%	67.7%	71.1%	0.766
PD	27.3%	29.0%	31.6%	0.819
Donor age (years) *	48.3 ± 18.3	47.4 ± 18.3	49.2 ± 18.4	0.674
Type of donor	-	-	-	0.571
Deceased donor	96.1%	94.9%	97.4%	
Living donor	3.9%	5.1%	2.6%	
Cause of death (deceased donor)	-	-	-	0.152
DBD	97.3%	100.0%	94.6%	
DCD	2.7%	0.0%	5.4%	
Cold ischemia time (hours) ^	20.0 (18.0–24.0)	20.0 (16.0–23.0)	20.0 (20.0–24.0)	0.306
Duration of the first graft (years) ^	10.0 (6.0–14.0)	11.0 (6.5–15.0)	9.0 (5.7–14.2)	0.017
Cause of graft loss	-	-	-	0.105
Chronic allograft nephropathy	75.3%	87.2%	63.2%	
Antibody-mediated rejection	1.3%	2.6%	0.0%	
Recurrence	3.9%	2.6%	5.3%	
Arterial/venous thrombosis	14.3%	5.2%	23.7%	
Others	5.2%	2.4%	7.8%	
Time from graft failure to subsequent serum (months) ^	7.6 (5.9–12.5)	6.8 (5.8–10.2)	8.5 (6.7–17.2)	0.053
Graft nephrectomy	23.0%	5.3%	41.7%	<0.001
Blood transfusions from transplant failure to subsequent serum (≥1 RBCC)	49.4%	53.8%	44.7%	0.424
cPRA before transplant failure ^	0.0 (0.0–0.0)	0.0 (0.0–21.4)	0.0 (0.0–0.0)	0.433
Group 1 (“No-CNI 3mo”)	44.2%	23.1%	65.8%	<0.001
Group A (“No-CNI 6mo”)	53.4%	31.6%	77.1%	<0.001

**Table 6 biomedicines-08-00072-t006:** Results of the multivariate logistic regression analysis for delta cPRA > 0%. The bold: *p* values less than 0.05 defined statistical significance

Multivariate logistic regression analysis for delta cPRA > 0%	**OR**	**95% CI**		***p***
		**INF**	**SUP**	
Graft nephrectomy	10.8	1.6	70.0	**0.012**
Group 1 (“No-CNI 3mo”)	5.4	1.7	17.1	**0.004**
Recipient age	1.0	0.9	1.0	0.449
Duration of the first graft	1.0	0.9	1.1	0.570
	**OR**	**95% CI**		***p***
		**INF**	**SUP**	
Graft nephrectomy	9.8	1.5	63.2	**0.016**
Group A (“No-CNI 6mo”)	5.6	1.7	18.3	**0.004**
Recipient age	1.0	0.9	1.0	0.709
Duration of the first graft	1.0	0.9	1.1	0.692

**Table 7 biomedicines-08-00072-t007:** Patient characteristics in relation to the development or not of de novo DSA (dnDSA) after transplant failure. Continuous variables were expressed as mean ± SD (*) or median and interquartile range (^) according to their distribution. The bold: *p* values less than 0.05 defined statistical significance.

dnDSA and Different Variables	*n* = 73	No dnDSA *n* = 40	dnDSA *n* = 33	*p*
Recipient age (years) *	60.1 ± 13.2	60.2 ± 13.3	59.9 ± 13.3	0.907
Recipient sex (male)	65.8%	65.0%	66.7%	0.881
Cause of CKD:	-	-	-	0.138
Vascular	6.8%	7.5%	6.1%	
Diabetes	15.1%	7.5%	24.2%	
Others	78.1%	85.0%	69.7%	
HTN	94.5%	97.5%	90.9%	0.218
Diabetes	32.9%	31.4%	43.3%	0.321
RRT before first transplant:	-	-	-	-
HD	63.0%	65.6%	75.8%	0.369
PD	26.0%	31.3%	27.3%	0.724
Donor age (years) *	48.1 ± 18.2	46.9 ± 18.9	49.6 ± 17.6	0.535
Type of donor	-	-	-	0.108
Deceased donor	95.9%	92.5%	100.0%	
Living donor	4.1%	7.5%	0.0%	
Cause of death (deceased donor)	-	-	-	0.129
DBD	97.1%	100.0%	93.9%	
DCD	2.9%	0.0%	6.1%	
Cold ischemia time (hours) ^	20.0 (18.0–24.0)	20.0 (18.0–24.0)	20.0 (17.0–24.5)	0.504
Duration of the first graft (years) ^	10.0 (6.0–14.0)	11.0 (6.5–14.5)	8.0 (5.7–11.7)	0.007
Cause of graft loss	-	-	-	0.129
Chronic allograft nephropathy	74.0%	85.0%	60.6%	
Antibody-mediated rejection	1.4%	0.0%	3.0%	
Recurrence	4.1%	5.0%	3.0%	
Arterial/venous thrombosis	15.1%	7.5%	24.2%	
Others	5.4%	2.5%	9.2%	
Time from graft failure to subsequent serum (months) ^	7.6 (6.0–12.1)	6.8 (5.8–8.5)	9.1 (6.7–18.4)	0.083
Graft nephrectomy	22.9%	8.1%	39.4%	0.002
Blood transfusions from transplant failure to subsequent serum (≥1 RBCC)	50.7%	55.0%	45.5%	0.417
cPRA before transplant failure ^	0.0 (0.0–0.0)	0.0 (0.0–0.0)	0.0 (0.0–0.0)	0.870
Group 1 (“No-CNI 3mo”)	42.5%	17.5%	72.7%	<0.001
Group A (“No-CNI 6mo”)	52.2%	23.1%	90.0%	<0.001

**Table 8 biomedicines-08-00072-t008:** Results of the multivariate logistic regression analysis for dnDSA. The bold: *p* values less than 0.05 defined statistical significance.

Multivariate logistic regression analysis for dnDSA	**OR**	**95% CI**		***p***
		**INF**	**SUP**	
Graft nephrectomy	3.4	0.5	20.4	0.170
Group 1 (“No-CNI 3mo”)	7.9	2.3	26.5	**0.001**
Recipient age	1.0	0.9	1.0	0.633
Duration of the first graft	0.9	0.8	1.0	0.662
	**OR**	**95% CI**		***p***
		**INF**	**SUP**	
Graft nephrectomy	2.6	0.3	19.5	0.341
Group A (“No-CNI 6mo”)	23.2	5.3	100.6	**<0.001**
Recipient age	1.0	0.9	1.0	0.893
Duration of the first graft	0.9	0.8	1.0	0.561

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
