# Peer review of "Immunosuppression with Calcineurin Inhibitor after Renal Transplant Failure Inhibits Allosensitization"

_biomedicines, 2020, doi:10.3390/biomedicines8040072_

Round 1

Reviewer 1 Report

Lopez del Moral Cuesta et al seek to answer an important biomedical question regarding pharmacologic ways to prevent renal allograft sensitization in patients who have suffered failure of a first transplant.  Their suggestion that maintenance CNI therapy should be continued post-transplantectomy or post-graft failure has been suggested in recent literature, which the authors cite here.  Regarding novelty of the findings, there are very few studies that have addressed this question to any degree, and therefore the contribution here is still important to add to the literature in this field.  The Introduction and Discussion sections are fairly good.

The Results section, and in general the specific questions asked and conclusions drawn must be expressed much more clearly throughout the paper.  This reviewer cannot fully evaluate the value of this project unless there is a substantial re-write in which the scientific method is clearly demonstrated.  There is no problem with English.  The problem is we are presented with giant tables full of numbers and little if any expression of each hypothesis, how it is addressed by specific data, and then conclusion.  The first four paragraphs in Results are essentially lists of different numbers.  Then three paragraphs present Tables 1-8 and several figures with very little explanation.  This is a cursory treatment.  If there are piles of numbers that contribute no interpretations, get rid of them, and explain every useful experimental question, how it is addressed, and what is concluded.  One example:  "Similarly, graft nephrectomy, group 1 and group A were associated with cPRA>75% after transplant failure, as shown in tables 3 and 4"; but why does Table 4 "graft nephrectomy" show no statistically significant result in top portion (p=0.053)?  If that is not the p-value the statement refers to, then why is that p-value there?  Please explain everything carefully.

cPRA:  explain everything about what this is, and explain what 75% cutoff means and why it is chosen.

delta cPRA:  A lot of data is shown for this, but it is not clear what questions are being asked, and what the answers are.  What did we learn from this?  If it is important, should that be included in the abstract?

Figure 1 -- this should be improved by better inclusion of all important populations in the study.  The biggest reported finding is about 6-month time period, which does not appear in Figure 1.  Group 1, Group A, etc., all that should be included in a clear manner.

What was learned from previously sensitized patients, versus unsensitized patients?

What was learned from all the other medicinal treatment groups?  Did corticosteroids have an effect?  Did they have no effect, and that is why it is acceptable to include both +/- in a single group?  What is the evidence that your conclusions about CNI treat it as an isolated variable, or does it only work if corticosteroids were also co-administered?

Finally, when the questions and answers are clear, the most important ones should be summarized in the abstract, and discussed in the paper.  Right now, the suggestion is that the field needs an IS regimen, but does this paper propose one?  Previous work has done that to some small degree (not enough, I agree with the present authors), but at least suggesting the 3 ng/mL dose.

This reviewer believes it is possible the current dataset and paper might contribute importantly to the field, but this is not yet clear.

Author Response

Dear reviewer, thank you for the recommendations and suggestions on the manuscript.

We have made modifications to the article, and we will respond to each suggestion proposed below.

  • We had previously summarized the results with the names of groups (A, 1 ...) instead of repeatedly writing the characteristics of each group (no-CNI at 3 months ...) We have explained the results better according to the tables. The modifications are underlined in yellow.
  • In the case of previously non-sensitized patients, transplantectomy is not significantly related to cPRA>75% (p> 0.05) in the multivariate analysis, and now we detail it in the text. The modifications are underlined in yellow.
  • cPRA provides an estimate of the percentage of deceased organ donors that will be crossmatch incompatible for a candidate, and now we specify this in the text since it is of great importance in our study. We set the limit of 75% since it is a round number, and it is very close to the median value (46.7% of patients have cPRA> 75%). Similarly, the value 0% was chosen to establish the limit and divide the patients into two groups in the case of deltacPRA. This is all specified now in the manuscript in Materials and Methods section (underlined in yellow).
  • Delta cPRA: this shows the difference between cPRA post-transplant failure and before transplant failure. Therefore, this value represents whether the patient has developed antibodies after graft loss. One of the strengths of our study is to have sera not only after graft loss, but also immediately prior to graft loss. This allows evaluating not only the overall percentage of cPRA, but the difference between both sera. Therefore, it provides additional information that we consider important and that enhances the result of our study (the maintenance of CNI at least until the sixth month after transplant failure prevents the development of anti-HLA antibodies). It is important additional information, but it is not in the abstract due to the word limit set in the publication. If necessary and considered important, the abstract could be modified to introduce this data.
  • We have modified Figure 1, adding the characteristics of groups A and B (No-CNI or CNI at 6 month). In addition, the characteristics of the patients belonging to these groups are explained (underlined in yellow).
  • Most patients in the study had cPRA 0% before graft loss. But some patients, as explained in the manuscript, were already sensitized. This does not affect the analysis of variables such as delta cPRA or de novo A small percentage of patients already had cPRA>75% before losing the graft, and therefore these patients should be eliminated for this analysis. For this reason, in the analysis of cPRA>75% after transplant failure, patients who were previously not sensitized were specifically analyzed, thus being able to interpret it correctly.
  • CNI and/or corticosteroids was the immunosuppression of choice in most patients who had treatment after transplant failure. Corticosteroid maintenance, in monotherapy or in combination with other immunosuppressive drugs, at the third and sixth month after graft loss was not significantly associated with cPRA≤75% in all patients and in previously non-sensitized patients. Similarly, the maintenance of corticosteroids at the third and sixth month was not associated with delta cPRA≤0% and no de novo DSA formation (p values ​​of significance are shown in the modified abstract, demonstrating the absence of a significant relationship between the maintenance of corticosteroids at the third and sixth month after graft failure with the different variables - cPRA> 75%, delta cPRA> 0% and dnDSA). For all the above and following the little described in the literature so far, our patients were classified according to the maintenance or not of CNI at the third (groups 1 and 2) or sixth month (groups A and B) after transplant failure. This is an important point to understand our analysis and it is now specified (underlined in yellow) in the manuscript.
  • The main conclusion of our study and our suggestion is to maintain CNI until at least the sixth month after transplant failure. This should be done when there is no longer a high antibody titer (then, there would be no benefit in the prevention of HLA sensitization). One of the limitations of our study, as described in the conclusions, is the absence of CNI levels in those patients who maintained it after graft loss. In an earlier study, commented in our manuscript, they recommend maintaining CNI to obtain levels ≥ 3 ng/mL. Due to the lack of a clear protocol of action, no CNI levels were extracted in our patients, and we cannot establish "protective levels" of CNI against HLA sensitization. More studies and prospective studies are needed, so they can demonstrate and establish an action pattern in the management of immunosuppression after transplant failure. We believe that our study supports the little described so far in the literature and may be the beginning for prospective studies and / or studies with greater number of patients and fewer limitations.

We have modified the manuscript and we have answered the proposed suggestions and doubts about the study. We send the modified manuscript and await your response.

Thank you in advance,

Covadonga López del Moral Cuesta et al.

Reviewer 2 Report

Del Moral Cuesta et al. present their study of a cohort of 77 renal transplant recipients with graft failure re-listed for retransplant and testing the association between ongoing maintenance CNI and the development of allosensitization, which may preclude the use of donors with those HLA antigens. The authors outline that there is equipoise on this point in the management of renal transplant recipients. They conclude that lack of CNI maintenance increases risk of high cPRA, increased cPRA compared to baseline and dnDSA, both in straight comparison and in multivariable analysis. Overall the analysis is reasonably well done and thorough and I suspect the results are robust, given they have been analyzed in several different ways and are in keeping with plausibility and prior work. There are noted inherent weaknesses in the design however, some of which the authors acknowledge, and which in totality may limit the study’s utility.

Major comments

  • As the authors outline in their introduction, these findings are not particularly novel, but rather are a useful adjunct to the existing literature
  • The number of omitted cases on the basis of missing data (sera for HLA antibodies) is a major potential source of bias, particularly in the context of a small study population such as this
  • The no-CNI arm is contaminated with people who were on immune suppression in other forms. Thankfully, this would be likely to bias the results towards null so the persistence of a positive finding is probably reassuring, but it should be acknowledged as a limitation.
  • The lack of any CNI levels is an odd omission and limits the generalizability and informativeness of the study
  • Should blood transfusion have been in the multivariable models as opposed to age? The lack of a simple comparative difference does not mean it is not a confounder, particularly given its known association with allosensitization. Why was age chosen over this?

Minor comments

  • The results section would be better if it was compartmentalized into sections with headings. As it stands, it’s a wall of text and bit difficult to digest, particularly with the analyses differing very slightly from section to section.
  • The introduction is overly long and I would suggest eliminating some non-essential information or moving it to the discussion

Author Response

Dear reviewer, we appreciate the assessment of our study and the proposed suggestions.

We have modified the manuscript, and then we respond to each suggestion:

  • We believe that our study provides evidence on the immunosuppression that should be maintained after transplant failure, as there is little evidence in the literature. Despite this, our study has several limitations, as explained in the manuscript. We have added the limitation that some of the patients in the no-CNI group (groups 1 and A) have other immunosuppressive drugs, but this probably enhances the result in favor of CNI in the prevention of sensitization after graft failure. This has been modified in the article (underlined in yellow).
  • In the multivariate analysis we have selected the variables that were significant in the average’s comparison or univariate analysis. Although the recipient age was not significantly related to any variable (cPRA>75%, delta cPRA>0%, dnDSA), age was added in the multivariate analysis as a "standard" variable, but if it is considered that way, it could be withdrawn from that analysis.
  • Blood transfusions may be a factor that promotes HLA sensitization. In our study, the number of transfusions from transplant failure to serum after graft loss was collected. In the average’s comparison, having received at least one blood transfusion after transplant failure was not significantly associated with cPRA>75%, delta cPRA>0% delta or dnDSA. For this reason, this variable was not included in the multivariate logistic regression analysis.
  • We have modified this, and we have performed the multivariate analysis by eliminating the variable "recipient age" and adding “blood transfusions”. This does not change the results, except for the risk of cPRA>75% in previously non-sensitized patients if there is no-CNI maintenance at 6 months after transplant failure (no-CNI at 6 months is associated with cPRA>75% in non-sensitized patients, but this is not statistically significantly). Interestingly, having received at least one blood transfusion in our study appears as a "protective factor" for antibody development in the multivariate analysis, although not significantly (contrary to what is described in the literature, since blood transfusions favor HLA sensitization. For this reason, we believe that perhaps blood transfusion data could be omitted from multivariate analysis).
  • Just in case, we have changed all the multivariate analyzes of our manuscript and we show it in the attached file (underlined in yellow). The parts of the article (and the abstract) that referred to multivariate analysis have been modified. We propose this new form of analysis including blood transfusions and eliminating the "age" variable, being a possible option if it is considered appropriate to include the variable "blood transfusions" in the multivariate logistic regression.
  • We have slightly shortened the introduction and classified the results into sections.

We have modified the manuscript and we have answered the proposed suggestions and doubts about the study. We send the modified manuscript and await your response.

Thank you in advance,

Covadonga López del Moral Cuesta et al.

Round 2

Reviewer 1 Report

The authors are to be thanked for the improvement of clarity in the revision of this interesting manuscript. Minor edits are now suggested to the authors, with final decisions made between the authors and editors:

(1) Title, change “prevents” to “inhibits”. 

(2) Abstract. The undefined abbreviations are not understood by a general Biomedicine audience. Please define each abbreviation in the abstract. Also, cPRA and DSA need in abstract a small clarifying phrase so the reader can understand. Something like, “It was possible to analyze the development of de novo donor specific antibodies (DSA) and calculated panel reactive antibodies (cPRA) where high values indicate allo-sensitization and incompatibility with potential second transplants.”

(3) In Abstract, to save words the authors might choose to omit the sentence about graft removal being related with high cPRA. The reason is this is not a title point of the paper, and there might be a good reason for this in the authors’ expertise. For example, the authors do not insist that surgeons should cease removing failed transplants, despite this observation. But the authors do suggest to keep failed organs in presence of CNI, so because the latter conclusion is favored and the former is not, perhaps the abstract should not be interrupted with that other observation.

(4) This reviewer feels that Figure 1 should start with the 147 patients, not the 253. This is because 253 is defined as “transplant failure”, but the 106 removed did NOT have transplant failure but rather died with functioning graft. It would be fair and more clear to start with 147 patients defined as “transplant failure”.

Author Response

Dear reviewer,

Thank you for the suggestions and the evaluation of our manuscript. Now we have made minor modifications:

  1. We have changed the title.
  2. We explain the abbreviations in the abstract, so the reader can now understand everything more easily.
  3. We have omitted the phrase in relation to transplantectomy in the abstract, giving more importance to our main conclusion (the absence of CNI favors HLA sensitization).
  4. Now figure 1 starts with 147 patients.

We send the modified abstract and we await your response.

Thank you in advance,

                                                                                                                 López del Moral Cuesta et al
